# Histone Modifying Enzymes in Gynaecological Cancers

**DOI:** 10.3390/cancers13040816

**Published:** 2021-02-16

**Authors:** Priya Ramarao-Milne, Olga Kondrashova, Sinead Barry, John D. Hooper, Jason S. Lee, Nicola Waddell

**Affiliations:** 1Medical Genomics Group, QIMR Berghofer Medical Research Institute, Brisbane, QLD 4006, Australia; priyaramaraomilne@gmail.com (P.R.-M.); Olga.Kondrashova@qimrberghofer.edu.au (O.K.); Nic.Waddell@qimrberghofer.edu.au (N.W.); 2Faculty of Medicine, The University of Queensland, Brisbane, QLD 4006, Australia; 3Department of Gynaecological Oncology, Mater Hospital Brisbane, Brisbane, QLD 4101, Australia; sinead.barry@mater.org.au; 4Mater Research Institute, The University of Queensland, Translational Research Institute, Woolloongabba, QLD 4102, Australia; john.hooper@mater.uq.edu.au; 5Epigenetics and Disease Laboratory, QIMR Berghofer Medical Research Institute, Brisbane, QLD 4006, Australia; 6School of Biomedical Sciences, Queensland University of Technology, Brisbane, QLD 4000, Australia

**Keywords:** gynaecological cancers, epigenetics, epigenetic enzymes, epigenetic modifiers, histone modifiers, epigenetic treatment

## Abstract

**Simple Summary:**

Epigenetics is a process that allows genetic control, without the involvement of sequence changes to DNA or genes. In cancer, epigenetics is a key event in tumour development that can alter the expression of cancer driver genes and result in genomic instability. Due to the critical role of epigenetics in malignant transformation, therapies that target these processes have been developed to treat cancer. Here, we provide a summary of the epigenetic changes that have been described in a variety of gynaecological cancers. We then highlight how these changes are being targeted in preclinical models and clinical trials for gynaecological cancers.

**Abstract:**

Genetic and epigenetic factors contribute to the development of cancer. Epigenetic dysregulation is common in gynaecological cancers and includes altered methylation at CpG islands in gene promoter regions, global demethylation that leads to genome instability and histone modifications. Histones are a major determinant of chromosomal conformation and stability, and unlike DNA methylation, which is generally associated with gene silencing, are amenable to post-translational modifications that induce facultative chromatin regions, or condensed transcriptionally silent regions that decondense resulting in global alteration of gene expression. In comparison, other components, crucial to the manipulation of chromatin dynamics, such as histone modifying enzymes, are not as well-studied. Inhibitors targeting DNA modifying enzymes, particularly histone modifying enzymes represent a potential cancer treatment. Due to the ability of epigenetic therapies to target multiple pathways simultaneously, tumours with complex mutational landscapes affected by multiple driver mutations may be most amenable to this type of inhibitor. Interrogation of the actionable landscape of different gynaecological cancer types has revealed that some patients have biomarkers which indicate potential sensitivity to epigenetic inhibitors. In this review we describe the role of epigenetics in gynaecological cancers and highlight how it may exploited for treatment.

## 1. Background

Cancer is a multifaceted group of diseases that develop due to an interplay between genetic and epigenetic factors. Somatic mutations alone do not account for the tumourigenic characteristics of cancer cells; and epigenetic deregulation of oncogenes and tumour suppressors is another important mechanism contributing to carcinogenesis [1,2]. There is a great interest in epigenetic drug targets in cancer [3,4], an area actively pursued in gynaecological cancers. Epigenetic inhibitors or “epidrugs” are being studied for their potential anti-tumour activity, and may be beneficial for cancers with global dysregulation of gene expression. The potential of these inhibitors to target reversible modifications to the genome, coupled with their ability to influence the expression of multiple genes concomitantly, make them attractive as novel anticancer compounds, through re-expression of tumour suppressor genes. This review will outline and discuss studies pertaining to epigenetic factors with a focus on histone modifying enzymes in common gynaecological cancers, specifically, ovarian, endometrial and cervical cancers. The mechanisms by which epigenetics contribute to tumorigenesis and the evidence that implicates epigenetic enzymes, in particular histone modifying enzymes, as treatment targets will be discussed.

### 1.1. DNA Methylation

DNA methylation, which has a key role in cancer [1,2], refers to the addition of methyl groups to DNA residues by groups of enzymes known as DNA methyltransferases (DNMTs) or demethylases. In particular, methylation of cytosine residues occurring in CpG dinucleotides can be methylated to form 5-methylcytosines. CpG sites are clustered within the genome to form CpG islands, with approximately 70% of genes in the human genome containing a CpG island within their promoter region. Global hypomethylation changes within the genome are associated with genomic instability, while hypermethylation occurs frequently at gene promoter regions. There is extensive evidence that hypermethylation of CpG sites in promoter regions is a crucial mechanism for tumour suppressor gene and microRNA deactivation [5,6], and can account for discrepancies in cases whereby gene mutations do not correlate with respective mRNA levels. 

### 1.2. Histone Modifications

In addition to CpG methylation of DNA, proteins within nucleosome subunits are also subject to covalent modifications. Nucleosomes are comprised of a core set of histones that exist as four dimers of histone H2A, H2B, H3 and H4. Histones are the major determinants of chromosomal conformation and stability, and are amenable to post-translational modifications. These modifications can induce changes in chromatin regions that may result in alteration of gene expression. Histone marks that signify active promoters generally include H3 lysine 4 trimethylation (H3K4me3) [7], while hallmarks of histones flanking inactive promoters are H3 lysine 27 trimethylation (H3K27me3) [8], as well as H3 lysine 9 trimethylation (H3K9me3) [9]. Additionally, there is a tight association between enhancers and regions consisting of H3 lysine 4 monomethylation (H3K4me1) [9]. Global overrepresentation of repressive histone marks is commonly found in cancers compared to normal tissue, and can result in the silencing of important tumour suppressor genes [10]. 

### 1.3. Enzymes Involved in Epigenetic Regulation 

Histone modifications are catalysed by histone modifying enzymes known as writers, erasers or readers. Writers are enzymes which catalyse addition of post-translational modifications onto histone tails, such as lysine methyltransferases (KMTs), histone acetyltransferases (HATs) and ubiquitin ligases. Erasers, such as histone deacetylases (HDACs) that are subdivided into four classes [11], as well as lysine demethylases (KDMs) and deubiquitinating enzymes remove post-translational modifications (Figure 1). Readers contain motifs that recognise post-translational modifications, recruiting co-factors, which modulate transcription. Together, these enzymes can modulate transcription, using a combination of histone marks to elicit responses by controlling the addition and removal of a variety of modifications, such as methylation, acetylation, ubiquitination, phosphorylation and sumoylation [8]. The same modification in a different position in the amino acid tail can recruit completely different chromatin remodelling complexes, and exhibit a total shift in function [12]. Collectively, varying combinations of covalent modifications comprise the “histone code” – a complex set of signals which recruit enzymes and alter the structure of chromatin, prompting conformational changes in nucleosomes [12]. The complexity of the histone code, in combination with other epigenetic factors allow for fine control of chromatin dynamics and therefore gene activation and repression. There is evidence that histone modifying enzymes are involved in tumorigenic behaviour of cancers including gynaecological cancers [13] (Table 1). 

While histone modifications can act alone to recruit co-factors, there is significant crosstalk between different types of epigenetic modifications. This involves cooperation between the enzymes that catalyse these modifications, such as DNMTs that transfer methyl groups onto CpG sites within promoters of genes to elicit gene silencing, and histone modifying enzymes that transfer covalent modifications, such as methyl and acetyl groups onto amino acid tails of histones. For example, DNMT1, which transfers methyl groups onto CpG islands, interacts with KMTs, contributing to H3K9 and H3K27 methylation [33], as well as several HDACs [33]. In addition, EHMT2 (commonly known as G9a) and EZH2, which are the methyltransferases of H3K9me2 and H3K27me3, respectively, also interact with each other [34]. G9a can modulate recruitment of the Polycomb Repressive Group 2 (PRC2) complex, which contains EZH2 as the enzymatic domain, to catalyse H3K27me3. Together, G9a and EZH2 co-repress target genes, although the finer mechanisms remain unclear [34]. The activity of both enzymes increases in hypoxic tumour microenvironments and is linked to the silencing of tumour suppressor genes and survival in breast cancer [35], while EZH2 transcription is enhanced by HIF-1α to promote more aggressive phenotypes [36]. As such, chromatin remodelling enzymes may form a network of interactions which are involved in multiple cellular processes within a variety of cancers.

## 2. Epigenetic Modifiers in Ovarian Cancer

Ovarian cancer has a poor prognosis, with a 5-year survival rate of 40–50% [37]. It is a heterogeneous disease comprised of multiple subtypes, the most common being epithelial ovarian cancer (EOC), which accounts for > 90% of all cases. The vast majority of cases are an aggressive and rapidly progressive subtype, high grade serous ovarian carcinoma (HGSOC) [38]. Other rarer subtypes include low-grade serous, endometrioid, mucinous and clear cell carcinomas [38]. Gene expression subtypes of EOC have been described with clinical significance [39], which have also been replicated using methylation profiling [40]. The genomic landscape of HGSOC has been profiled by TCGA [41] and ICGC [42] revealing that 95% harbour *TP53* somatic mutations. Other somatic genomic features include widespread genomic rearrangements including *CCNE1* amplifications in around 20% of cases [43], and recurrent gene breakage of *RB1*, *NF1*, *RAD51B* and *PTEN* [42]. Other key HGSOC genes are involved in homologous recombination (HR) repair, with up to 18% of patients carrying a germline *BRCA1* and *BRCA2* mutation [44], an additional 8% of cases containing somatic *BRCA1/2* mutations [41,45], and around 10–15% of cases have promoter hypermethylation of *BRCA1*, which leads to BRCA1 silencing [46,47]. Other aberrations in HR genes, include mutations in genes such as *PALB2, BRIP1* and *RAD51C* [48], and methylation of *RAD51C* [49]. 

Drug development for ovarian cancer has generally lagged behind advances for other solid malignancies. However, recently PARP inhibition [50] is beginning to transform the treatment landscape for HGSOC cases. Multiple Phase III trials have supported the use of PARP inhibitors [51], especially those with germline or somatic *BRCA1/2* variants or those with a defective HR repair [52], and these agents are now mainstays for recurrent disease with significant potential, also as maintenance therapies after primary treatments. Targeted therapies such as erlotinib for non-small cell lung cancer [53] and Herceptin have become entrenched into treatment regimens for breast cancer [54], but attempts to recapitulate such success in gynaecological cancers has been ineffective thus far. A significant impediment to the development of targeted therapies in ovarian cancer is the presence of extensive intra- and inter-tumoural heterogeneity, and apart from HR deficiency, the lack of clinically actionable mutations. Therefore, targeting epigenetic enzymes in cancers that exhibit extensive dysregulation of gene expression may be a suitable alternative.

### 2.1. Histone Methyltransferases in Ovarian Cancer

Extensive evidence implicates KMTs in cancer. EZH2 is involved in ovarian cancer although multifaceted roles likely exist, so whether it is acting as a tumour suppressor or oncogene is unclear and may be context dependent. For example, inhibition of EZH2 promotes epithelial-to-mesenchymal transition in ovarian cancer cells [14]. However, knockdown of EZH2 levels also correlate with an induction of apoptosis in epithelial ovarian cancer cells, and regression of tumour xenografts [55]. Similarly inhibition of KDM6B, an H3K27me3 demethylase, may also induce apoptosis in ovarian cancer cells [16]. The presence or absence of EZH2 has potential treatment implications. Knockdown of EZH2 was able to re-sensitise the A2780-DDP cell line that is resistant to platinum through G2/M cycle arrest [56], which may have important implications for patients with platinum-resistant ovarian cancer. While overexpression of *EZH2* inhibits phosphorylation of the BRCA1 protein at serine 1423, reducing its function [57], whether this would result in HR deficiency and subsequent response to PARP inhibition is not known. Co-inhibition of EZH2 and DNMT1 in a tumour may have the potential to reprogram the immune microenvironment and increase the efficacy of PD-L1 checkpoint blockade therapy [58]. This suggests a role in modulating immune responses, and that combined methyltransferase and immune checkpoint inhibition may enhance anti-tumour activity. 

G9a is a methyltransferase that can catalyse mono- and di-methylation of H3K9. Elevated expression of G9a has been reported in a cohort of ovarian cancer patients, and was significantly associated with poor prognosis [17]. G9a has also been associated with metastases, as omental, peritoneal and lymph node metastases showed significantly elevated levels of G9a compared to matched primary tumours [17] and overexpression of G9a increased migration of cells in in-vitro scratch wound assays [17]. Another methyltransferase with an emerging role in cancer progression is DOT1L, which mediates catalysis of mono-, di- and tri-methylation of H3K79. Most research has investigated its role in mixed lineage leukaemia [59,60,61]. However, DOT1L also modulates ERα target genes [62], inducing chemoresistance [63] with its elevated expression predicting poor prognosis in ovarian cancer patients [64]. 

The histone methyltransferase SMYD2 (known as KMT3C), in addition to catalysing H3K4me3, also methylates PARP1 at lysine 528 which increases the activity [65]. PARP1 is involved in DNA damage repair [66], and its elevated activity is a contributor to survival and chemoresistance [67]. Increased staining of SMYD2 has been observed in ovarian and cervical cancers, however its involvement in ovarian cancer in terms of histone methyltransferase activity, or its ability to methylate non-histone proteins such as PARP1, remains unclear [68]. Furthermore, whether tumours with a high level of SMYD2 would respond to PARP inhibition is not known. 

### 2.2. Histone Demethylases in Ovarian Cancer

Members of the KDM family also have known roles in cancer. KDM1A overexpression has been observed in a variety of different malignancies, including prostate [69] and bladder [70] cancers and it may have a role in promoting tumorigenesis and metastasis. Activation of the PI3K pathway in several cancers is well known and is involved in the proliferation and migration of ovarian cancer cells. The ligand upstream of the P13K pathway, EGF, upregulates KDM1A, which results in decreased global H3K4me2 levels [18]. In support of a role for KDM1A in cell migration, its overexpression in several ovarian cancer cell lines increases cell motility [18]. While inhibition of KDM1A activity with tranylcypromine decreases migration of the SKOV3 ovarian cancer cell line, and restores global expression of H3K4me2 [18].

A key hallmark of HGSOC is loss of p53 and widespread chromosomal instability due to genomic rearrangements. Overexpression of the demethylase KDM4A induces specific acquisition of copy gains at 1q12, 1q21, and Xq13.1 [25], suggesting that this enzyme and histone modifying enzymes may be contributing to genomic instability. Another KDM, KDM3A, is overexpressed in various ovarian cancer tissues, and its expression was significantly elevated in three cisplatin resistant ovarian cancer cell lines, compared to normal ovarian tissue [26]. Knockdown of KDM3A reduced proliferation in vitro and tumour growth in vivo [26], and restored cisplatin sensitivity in resistant cell lines via apoptosis induction [26]. Given the role of KMTs and KDMs in ovarian cancer and their potential treatment implications, further investigation is required to clarify the mechanisms involved.

### 2.3. HDACs in Ovarian Cancer

Normal ovarian surface epithelium exhibits weak nuclear expression of *HDAC1*, *HDAC2* and *HDAC3* [71], whereas elevated levels are reported in a variety of ovarian tumour types [72,73]. High expression of *HDAC1* has been linked to poor prognosis in endometrioid subtypes of ovarian and endometrial carcinomas [74]. There is mounting evidence supporting the inhibition of HDACs in cancer treatment. Drug-induced inactivation or gene silencing of HDAC1 suppressed ovarian cancer cell growth [75], and inhibition of HDAC1/2/3 using the inhibitor Panobinostat, slowed the growth of ovarian cancer in a xenograft mouse model [76]. Inhibition of HDAC2 in a chemosensitive ovarian cancer cell line, PEO1, increased the efficacy of carboplatin treatment and increased γH2AX foci, and caused downregulation of phosphorylated BRCA1 [77].

Chemoresistant relapses are a frequent occurrence in patients with HGSOC. HDACs have been implicated in chemoresistance and their inhibition can prolong chemosensitivity or induce sensitivity in inherently chemoresistant tumours. Treatment with DNMT inhibitors and HDAC inhibitors (HDACis) has been shown to re-sensitise the cells to platinum treatment [78]. Chemotherapy resistance is also associated with the presence of cancer stem cells (CSCs), which are thought to exist as a pluripotent fraction within a tumour harbouring multiple pro-survival characteristics that allow them to evade chemotherapy-induced death [79]. As such, there is a particular interest in eliminating this population of cells, in order to maintain, or induce chemosensitivity. *HDAC1* and *HDAC7* are involved in the generation and maintenance of these CSC populations in breast and ovarian cancer cell lines [80]. Encouragingly, treatment with a HDAC1 inhibitor dampened the CSC phenotype, reducing their tumorigenic characteristics [80]. HDACis can also affect DNMT1 levels, a DNA methyltransferase, and abolish the activity of DNMT1 by inhibition of HDAC2, thus depleting CpG island methylation at the promoters of tumour suppressor genes [81]. However such an effect could have significant negative results for patients that are HR deficient, as the loss of methylation from *BRCA1* or *RAD51C* gene promoter, would restore HR proficiency, rendering patients insensitive to PARP inhibition [82]. Therefore, the molecular makeup of the tumour should be considered prior to treatment selection. 

## 3. Epigenetic Modifiers in Endometrial Cancer 

Endometrial cancer is the fifth most prevalent cancer in women in the developed world [38]. While the overall survival for endometrial cancer is around 85%, the prognosis for locally advanced and metastatic disease is considerably worse, with the five-year survival rate dropping to 25% [37]. Endometrial cancer is characterised by four molecular subtypes with distinct treatment and survival outcomes. These subtypes were first described by the TCGA [83] and have since been validated by multiple groups [84,85]. These subtypes include *TP53*-mutant (also known as copy number high), mismatch repair deficient (MMRd; or microsatellite unstable), *POLE*-mutant and *TP53* wild-type (or copy number low). MMRd endometrial cancer is often caused by germline mutations in MMR genes, including *MSH2*, *MLH1* and *MSH6,* collectively recognised as Lynch syndrome [86]. *TP53*-mutant and MMRd molecular subtypes are known as the more aggressive subtypes associated with worse survival [85], so there is a strong clinical need for effective treatments, particularly for these subtypes and advanced stage disease. 

### 3.1. Histone Methyltransferases in Endometrial Cancer

Endometrial cancer cells have higher levels of G9a localised to the nuclei compared to normal tissues, which exhibit only weak nuclear staining [15]. Stronger G9a staining was correlated with depth of myometrial invasion [15], which is a strong predictor of poor prognosis and survival. Increased *EZH2* expression correlates with significantly lower survival rates in endometrial cancer [87]. Similar to ovarian cancer, G9a knockdown significantly attenuated migration of endometrial cancer cell lines [15]. Intriguingly, it also showed significantly diminished occupancy of DNMT1 at the promoter of the E-cadherin gene, *CDH1*, coupled with increased protein levels of E-cadherin protein, thus providing a potential mechanism by which G9a regulates gene expression and directly contributes to tumorigenic behaviour [15]. This may suggest that DNA methylation inhibition alone in this context would be ineffective at re-expressing E-cadherin, and removal of repressive H3K9me/H3K9me2 must also occur, highlighting the crosstalk of and complexity of epigenetic factors associated with the tumorigenic behaviour of cancer cells. 

### 3.2. Histone Demethylases in Endometrial Cancer 

Expression of KDM4B and KDM4A is higher in endometrial cancer tissue compared to normal endometrium tissue [28]. KDM4B is a histone demethylase which recognises and removes methyl marks from H3K9me2 and H3K9me3 to subsequently activate transcription. One study exploring the role of androgen receptor (AC) in endometrial cancer suggested that KDM4B together with AR can activate the well-recognised oncogene, *MYC,* by removing H3K9me3 marks in endometrial cancer cells with high basal levels of AR [28]. This effect was not observed in endometrial cancer cell lines with low levels of AR; instead, another histone demethylase, KDM4A, reduced levels of H3K4me3 methylation, an activation mark, at the promoter of the tumour suppressor gene p27kip1 [28]. These results suggested that both KDM4A and KDM4B together with AR have a role in endometrial cancer development and progression.

Depending on the context, KDM1A can act as a co-repressor or co-activator, catalysing either H3K9me1 or H3K4me2 [88]. This is achieved by exchanging transcriptional modules which enables the switch in activity. Association of KDM1A with the CoREST complex permits it to catalyse H3K4me2 demethylation [89], however, KDM1A is also responsible for the regulation of AR target genes. A study showed that inhibition of KDM1A via HCI2509 increased levels of H3K4me2 and H3K9me1, as well as increased global H3K27me3, proposing potential involvement in crosstalk with EZH2 and its modifications [19]. 

Taken together this shows that histone demethylases from the same family can potentially target different histone marks, resulting in oncogenic activation or suppressive mechanisms.

### 3.3. HDACs in Endometrial Cancer 

Similar to ovarian carcinomas, high-grade endometrial carcinomas express high levels of *HDAC1*, *HDAC2* and *HDAC3* [74], while less aggressive subtypes show lower levels of HDAC expression. In agreement with this, samples with high expression of HDACs showed a higher proliferating capacity in endometrial and ovarian cancer [74], consistent with the observation that HDACis can induce apoptosis and cell cycle arrest in vitro. 

## 4. Epigenetic Modifiers in Cervical Cancer 

With the introduction of a vaccine, cervical cancer is now a largely preventable disease, however it remains one of the most commonly diagnosed cancers in developing countries [90]. The main risk factor for cervical cancer is infection with variants of the human papilloma virus (HPV), which underlie the cause of almost all cervical cancer cases [91]. HPV infection alone is not sufficient for the development of cervical cancer; another event such as activation of oncogenes or deactivation of tumour suppressor genes is required to initiate carcinogenesis [92]. HPV contains an 8-kb circular genome which encodes for a number of proteins including viral “oncoproteins” E6 and E7 [91] which are responsible for the repression of host tumour suppressor genes *TP53* and *RB1*. Increased viral E7 inhibits binding of HDACs to hypoxia inducible factor 1 (HIF-1), thus activating transcription of pro-angiogenic genes downstream of HIF-1 [93]. Upon integration of the virus into the host genome, regulators of E6 and E7 by HPV proteins E1 and E2, are disrupted and there is a loss of transcriptional control of E6 and E7 [94]. Epigenetics also plays an important role in the modulation of these proteins, and may contribute to cervical cancer pathogenesis [95]. A screen to identify E2 binding partners found that it binds EP400, a component of the histone acetyltransferase complex NuA4/TIP60, as well as KDM5C [20]. EP400 also acts as a transcriptional corepressor, and KDM5C possesses demethylase activity against the active promoter marks, H34me2 and H3K4me3. This has led to the speculation that the transcriptional repressor activity of EP400 and KDM5C may aid in E2 mediated repression of the E6 and E7 oncoproteins [20]. 

## 5. Epigenetic Treatments in Gynaecological Cancers

Research into epigenetic regulation of cancers has facilitated the development of multiple inhibitors targeting DNA modifying enzymes. The development of DNA methyltransferase inhibitors and HDAC inhibitors (Figure 1), has fuelled the application of epigenetic treatments in gynaecological cancers. There have been a variety of preclinical studies testing epigenetic inhibitors in models of gynaecological cancers, some of which are highlighted in this review and Table 2, but this is not an exhaustive list as only representative examples of recent studies have been included. 

### 5.1. HDAC Inhibitors 

Of all histone modifying enzymes, HDACs have been most extensively studied for their potential as cancer therapies. An array of HDACis are currently being tested within a variety of cancer types, which has facilitated the refinement of these agents. Several HDACis are FDA approved and have shown potential in haematological cancers, however less so as a single agent in solid tumours, where they produce at most modest decreases in tumour growth. Instead, inhibiting histone modifying enzymes could show more promise as drug resistance modulators, as they may be able to induce chemosensitivity or drastically lower chemotherapy doses. Here, the impact of the development of these inhibitors is discussed, with an emphasis on gynaecological cancers.

A variety of epigenetic inhibitors in particular HDAC inhibitors are currently or have been trialled within Phase I or II trials for gynaecological cancers, a summary of trials included on ClinicalTrials.gov (accessed on 18 January 2021) is provided in Table 3. 

Although HDAC inhibitors have shown promise in T cell lymphomas [106], as well as other haematological malignancies [107], their efficacy in gynaecological cancers has been more limited. In 2008, a Phase II trial from the Gynecologic Oncology Group (GOG) showed that monotherapy of an oral dose of Vorinostat in patients with recurrent epithelial ovarian cancer, who were resistant to platinum based therapy, failed to produce encouraging results [108]. Vorinostat was also trialled as a combination treatment with carboplatin and gemcitabine in women with recurrent, platinum sensitive, fallopian tube or peritoneal cancer [109]. Although patients did demonstrate a response via the RECIST criteria, the study had to be terminated in its early stages due to toxicity. Despite many adjustments in dosing schedules, patients exhibited extensive haematologic toxicities with the incidence of Grade 4 thrombocytopaenia increasing to 23% with the addition of Vorinostat, from the 5% expected with gemcitabine and carboplatin alone. It would be interesting to determine if there is value of utilising Vorinostat in between chemotherapeutic regimens, or at a maintenance dose to achieve stable disease, or potentially with other agents which do not exacerbate side effects. 

Panobinostat is another HDACi that inhibits Class I, II and IV HDAC enzymes. A Phase III trial of Panobinostat in combination with bortezomib and dexamethasone (the PANORAMA 1 trial) in patients with relapsed multiple myeloma yielded a modest survival benefit of approximately 3 months median progression free of survival [110]. However, the side effect profile showed high level toxicity in patients receiving this treatment, potentially limiting its use in the clinic [111]. Even so, several in vitro studies have continued to explore the use of Panobinostat in gynaecological cancers. In HR-proficient ovarian cancer cell lines, Panobinostat downregulated DNA damage repair genes and induced sensitivity to the PARP inhibitor olaparib [112]. This potential synergistic effect of HDACis warrants further investigation to determine if HR proficient tumours can be sensitised to PARP inhibition. Interestingly, Panobinostat may also be able to impart re-sensitisation of platinum-based compounds as demonstrated in cisplatin resistant ovarian cancer cell lines [113], possibly by rewiring these cells to a BRCA-deficiency like phenotype. 

Another HDACi that has been trialled for treatment of ovarian cancer is Belinostat (Table 3). A Phase II trial of Belinostat alone in micropapillary/borderline and epithelial ovarian cancer found that it was generally well-tolerated with no Grade 4 toxicity events [114]. However there was only a modest progression-free survival improvement in the micropapillary/borderline cancer patients [114]. Belinostat was also tested in combination with carboplatin and paclitaxel [115], but the study was halted after the first stage due to little activity in the platinum-resistant ovarian cancers.

In the cervical cancer cell lines, SiHa and HeLa, the HDACi Panobinostat induces production of reactive oxygen species (ROS) and synergises strongly with topoisomerase inhibitors [116]. Specifically, Panobinostat induces cell cycle arrest and increases the percentage of cells in G1 cell cycle by affecting mitochondrial membrane potential, and increasing ROS. An increase in p21 was observed, consistent with the inhibition of CDK, illustrating the ability of Panobinostat to induce cell cycle arrest in the two tested cell lines. Significant downregulation of Bcl-xL, a component of mitochondrial anti-apoptotic machinery, was observed, in addition subsequent release of cytochrome c, a known precedent to apoptosis. 

Despite in vitro data supporting a role for HDACis in gynaecological cancers, the unfavourable side effect profile and lack of clear efficacy hampers their use in the clinic [111]. Additionally, most of the HDACis tested are pan-HDAC inhibitors which may have many off-target effects and may be contributing to the adverse side effects observed. Therefore, further study to develop more specific inhibitors with on-target activity may be warranted. 

### 5.2. Histone Methyltransferase and Demethylase Inhibitors

While HDACis are being extensively tested in clinical trials, progress in the development of histone methyltransferase inhibitors has been slower. Nonetheless, inhibitors targeting EZH2, KDM1A and DOT1L are being developed, while only inhibitors of EZH2 are currently in clinical trials for patients with gynaecological malignancies. 

Through high-throughput drug screening and optimisation, small molecule inhibitors to EZH2 have been developed [117], many of which are competitive for the methyl co-factor *S*-adenosyl-L-methionine (SAM) required for the enzymatic activity of EZH2. For example, GSK126 was administered to patients with diffuse large B cell lymphoma (DLBCL) in a clinical trial (NCT02082977) [118], although this study was terminated in 2017 due to lack of clinical response. However, a new EZH2 inhibitor with greater oral bioavailability and specificity, tazemetostat was approved by the FDA for adult patients with relapsed or refractory follicular lymphoma whose tumors are positive for an EZH2 mutation who have received at least 2 prior systemic therapies, and for those patients with no satisfactory alternative treatment options (Study E7438-G000-101, NCT01897571 [119]).

In gynaecological cancers there are several ongoing trials for patients with small cell carcinoma of the ovary (NCT03874455), endometrial and ovarian carcinoma (NCT03348631). Other clinical trials include combining tazemetostat with immunotherapy in treating patients with locally advanced or metastatic urothelial carcinoma (NCT03854474). It would be interesting to see whether this newer compound will show greater efficacy.

While it is unlikely that histone modifiers will be introduced to chemo-naïve patients, or patients with chemoresistant disease as a first-line therapy, their use in addition to conventional chemotherapeutic regimes has shown some promise pre-clinically. Discrepancies between in vitro data and clinical data need to be addressed if histone methyltransferase inhibitors are to progress to the clinic for patients with gynaecological malignancies. 

### 5.3. Targeting “Readers” of Histone Modifications

Another class of emerging anticancer agents are bromodomain and extra-terminal (BET) protein inhibitors that target the BET family proteins epigenetic “readers”, which recognise post-translational modifications such as those on histone residues. BRD4 is the most extensively studied member of the BET family of proteins; and binds acetylated lysines on histones. The relevance of BRD4 in cancer pertains to its occupancy at super enhancers of oncogenes, such as c-myc [120]. In ovarian cancer, preclinical models have suggested efficacy for BET inhibitors, and revealed a mechanism of action via disruption of Forkhead box protein M1 (FoxM1) pathway [32], one of the drivers of ovarian carcinoma. It has been subsequently shown in multiple studies that BET inhibitors can reduce HR activity and sensitize HR-proficient cell lines to PARP inhibitors [121,122,123,124,125]. This is significant as BET inhibitors may enable ovarian cancer patients that are HR-proficient access to PARP inhibition. 

Despite promising pre-clinical studies, clinical trials involving BET inhibitors have yielded mixed results. Many of these studies have been conducted in non-gynaecological cancers. For example, a dose finding clinical trial administering OTX015 to glioblastoma patients (NCT02296476) was established based on promising anti-tumour effects in preclinical glioblastoma models [126]. However, the trial was terminated after one year due to a lack of clinical activity. Similarly, a Phase I trial of OTX015 inadvanced solid malignancies revealed partial responses in some patients, but most patients suffered adverse effects, including grade 3 and 4 thrombocytopaenia [127]. At present, there are several clinical trials listed using BET inhibitors for various malignancies, although 9 have either been withdrawn or terminated. 

Several Phase I trials have been initiated with new BET inhibitors for a variety of tumour types including BMS-986158 for the treatment of recurrent or refractory solid tumors (CNS or Lymphoma) (NCT03936465 and NCT02419417), CPI-0610 for myelofibrosis (NCT02158858) and PLX51107 and azacitidine in treating patients with acute myeloid leukemia or myelodysplastic syndrome (NCT04022785). Two trials have entered Phase 2 examining the efficacy of the pan-BET inhibitor ZEN-3694 in combination with enzalutamide in metastatic castration-resistant prostate cancer patients (NCT04471974) or in combination with talazoparib in triple negative breast cancer (NCT03901469). In ovarian cancer, various pre-clinical studies have exhibited effective anti-tumour responses in vitro and in vivo [128,129,130], including strong synergistic effects when combined with PARP inhibitors [121,122,123,125,131]. Given the strong pre-clinical evidence for BET inhibitors in myc-addicted tumours, further study is warranted into this class of epigenetic inhibitor, and efforts should be focused into minimising the severe adverse effects encountered by patients undergoing BET inhibitor treatment.

### 5.4. Combination Therapy

The rationale for combination therapies in cancer treatment has been reviewed [132]. As alluded to throughout this review, combination therapy with various agents may augment the anticancer effect of epigenetic inhibitors [133,134]. For example, the combination of HDAC inhibitors and immunotherapy appears to be beneficial through transcriptional induction of endogenous retroviruses and testis antigens, resulting in a response called viral mimicry [133]. In vitro and in vivo xenograft studies in ovarian cancer cell lines and mouse models have shown that HDAC inhibitor treatment results in re-sensitisation of ovarian cancer cells to paclitaxel [135]. Additionally, the combination of PARP inhibitors with BET inhibitors resulted in increased DNA damage in ovarian cancer cells, potentially by blocking homologous recombination repair [136]. Moreover, epigenetic inhibitors can be combined with each other to induce cell death; for example, the combination of HDAC inhibition and demethylating agents such as 5-azacytidine and decitabine exerted synergistic effects in inhibiting proliferation of ovarian cancer cells [137]. The results from these studies illustrate the range of processes which can be modulated by epigenetic inhibition, and thus can be leveraged to enhance the cytotoxic activity of other agents. As such, more research into combination therapies with epigenetic inhibitors is well-warranted.

## 6. Concluding Remarks

Clearly, there is an urgent need for the development of novel therapies for metastatic disease in gynaecological cancers, alongside better strategies to achieve earlier diagnosis. Epigenetic enzymes play a vital role in carcinogenesis and may prove to be effective targets for cancer therapy. However, much progress needs to be made in order to ensure the right patients are targeted and the right combination of therapies is used to facilitate the translation of research findings into clinically relevant outcomes. There is evidence for the potential of targeting histone modifying enzymes in these aggressive cancers. However, rigorous preclinical studies, as well as careful drug development against robust biomarkers and well-designed clinical trials will be imperative if there is a chance for successful transition of inhibitors to histone modifying enzymes from the bench to the bedside. 

## Figures and Tables

**Figure 1 cancers-13-00816-f001:**
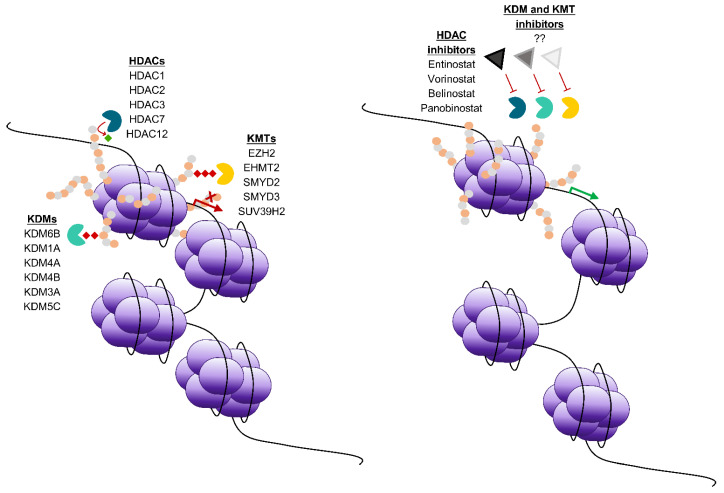
Histone modifying enzymes and epigenetic drugs in gynaecological cancers to restore the balance of epigenetic factors. Left panel; HDACs, KDMs and KMTs are often over-expressed across many gynaecological cancers. Mechanisms include addition of methyl groups and/or removal of acetyl groups at key histone tail residues, resulting in the repression of key tumour suppressor genes. Right panel; Epigenetic intervention via small molecule inhibitors to epigenetic enzymes, or “epidrugs” induces changes in chromatin configuration, resulting in re-expression of tumour suppressor genes. Examples of HDAC inhibitor names are provided, question marks denote that KDM/KMT inhibitors are to be determined. Green wedges represent lysine demethylases (KDMs), yellow wedges represent KMTs, blue wedges represent HDACs, red diamonds represent methyl groups, green diamonds represent acetyl groups, alternating grey and orange circles represent amino acid residues that comprise histone tails, large purple ovals represent histone subunits, triangles with black, dark grey and light grey borders represent HDAC inhibitors, KDM inhibitors and KMT inhibitors respectively.

**Table 1 cancers-13-00816-t001:** Histone modifying enzymes relevant to gynaecological cancers.

Enzyme	Enzyme Type	Mechanism	Modification	Cancer	Expression	Reference(s)
EZH2	Writer	Methyltransferase	H3K27me2, H3K27me3	Ovarian, Endometrial	↑	[14,15]
KDM6B	Eraser	Demethylase	H3K27me2, H3K27me3	Ovarian	↑	[16]
EHMT2	Writer	Methyltransferase	H3K9me, H3K9me2	Ovarian, Endometrial	↑	[15,17]
LSD1	Eraser	Demethylase	H3K4me2, H3K4me	Ovarian, Endometrial	↑	[18]
LSD1	Eraser	Demethylase	H3K9me	Endometrial	↑	[19]
EP400	Writer	Acetyltransferase	HAc	Cervical	↓	[20]
KDM5C	Eraser	Demethylase	H3K4me, H3K4me2, H3K4me3	Cervical	↓	[20]
SMYD2	Writer	Methyltransferase	H3K4me3	Ovarian	↑	[21]
SMYD3	Writer	Methyltransferase	H3K4me3, H4K5me	Cervical	↑	[21]
DOT1L	Writer	Methyltransferase	H3K79me2	Ovarian	↑	[22]
RNF20-RNF40 complex	Writer	Ubiquitinating enzyme	H2Bub	Ovarian	↓	[23,24]
KDM4A	Eraser	Demethylase	H3K9me3, H3K36	Ovarian	↑	[25]
KDM3A	Eraser	Demethylase	H3K9me, H3K9me2	Ovarian	↑	[26]
SUV39H2	Writer	Methyltransferase	H3K9me3	Cervical	↑	[27]
KDM4B	Eraser	Demethylase	H3K9me2, H3K9me3	Endometrial	↑	[28]
HDAC1	Eraser	Deacetylase	HAc	Ovarian, Endometrial	↑	[29,30]
HDAC2	Eraser	Deacetylase	HAc	Ovarian, Endometrial	↑	[29,30]
HDAC3	Eraser	Deacetylase	HAc	Ovarian, Endometrial	↑	[29,30]
HDAC7	Eraser	Deacetylase	HAc	Ovarian	↑	[31]
HDAC12	Eraser	Deacetylase	HAc	Ovarian	↑	[31]
BRD4	Reader	BET protein	-	Ovarian	↑	[32]

Up arrows (↑) represent increased expression in indicated cancer relative to normal tissue, down arrows (↓) represent decreased expression in indicated cancer relative to normal tissue.

**Table 2 cancers-13-00816-t002:** Recent examples of preclinical studies that have tested epigenetic inhibitors in gynaecological cancers.

Enzyme	Epigenetic Inhibitor	Combination Agent(s)	Cancer Type	Model/Cell Lines Tested	Inhibitor Dose and Treatment Duration	Outcome	Reference(s)
KMT4/DOT1L	EPZ004777	None	Ovarian	PEO1 and PEO4 cell lines	0.1 µM, 72 h	Growth arrest	[96]
EZH2	GSK126	Cisplatin	Ovarian	OVCAR3 and CA-MSC orthotopic mouse model	Various concentrations	Cell viability of OVCAR3 cells unaffected, but decreased ability of OVCAR3 cells to metastasise	[97]
EZH2	GSK126	5-AZA dC	Ovarian	NSG model	30 mg/kg, 3 times a week for 2 weeks	Increased efficacy of adoptive T-cell therapy in vivo	[58]
EZH2	GSK126	None, Cisplatin or Doxorubicin	Endometrial	Various cancer cell lines	0.025–20 µM, 8 days	Decreased cell proliferation and induction of apoptosis. Additive effects with cisplatin or doxorubicin	[98]
EZH2	DZNep	None	Cervical	HeLa and HeLa/DDP cells	Various concentrations, 72 h	Reversal of cisplatin resistance observed in the HeLa/DDP cell line	[99]
G9a (EHMT2)	BIX01294	None	Cervical	CaSki, HeLa and SiHa cell lines	5 µM, 72 h	Cell migration and invasion attenuated in BIX01294-treated cells	[100]
G9a (EHMT2)	BIX01294	None	Cervical	Subcutaneous SiHa cell line xenograft cervical cancer tumor model	5 mg/kg and 10 mg/kg, 39 days	Xenograft tumour growth significantly attenuated from day 29 at a dose of 10 mg/kg compared to control	[100]
G9a (EHMT2)	UNC0638	None	Ovarian	SKOV-3, ES-2, and PA-1 cell lines	2 µM, 48 h	Increase in metastasis suppressor genes such as CDH10	[101]
G9a (EHMT2)	UNC0638	None	Ovarian	SKOV-3 cell line	2 µM, 48 h	Decreased metastasis-related signaling	[17]
SUV39H1/SUV39H2	Chaetocin	None	Ovarian	OVCAR3 cell line	IC50–60.66 nM, 24 h	Inhibited proliferation, induced ROS accumulation and resulted in caspase-induced cell death in OVCAR-3 cells	[102]
SUV39H1/SUV39H2	Chaetocin	None	Cervical	HeLa and CaSki cell lines	150 nM, 24 h	Restored the innate immune response to exogenous DNA	[103]
KDM6A/6B	GSK-J4	None	Cervical	SiHa and HeLa cell lines	25–100 µM, 72 h	Decreased cell viability in SiHa cell line, no effect in HeLa cell line	[104]
KDM6A/6B	GSK-J4	None	Cervical	CaSki cell line	0–30 µM, 72 h	Decreased cell viability	[104]
KDM6A/6B	GSK-J4	None	Ovarian	A2780 cancer stem cell like cells	0.5–10 µM, 72 h	Decreased cell viability	[16]
LSD1	SP-2577	None	Ovarian	SWI/SNF-mutated cell lines	0.01–1.1 µM, 72 h	Affects cell viability, and induces expression of inflammatory cytokines in organoids	[105]
LSD1	HCI2509	None	Endometrial	AN3CA and KLE cell lines	IC50–500 nM, 96 h	Apoptotic cell death in cell lines, tumour regression in orthotopic xenografts	[19]
BRD4	JQ1, I-BET151	None	Ovarian	Various cell lines	0.01–10 µM	Cell cycle arrest in all subtypes of ovarian cancer cell lines tested	[32]
BRD4	JQ1	None	Ovarian	OVCAR-3 cell line xenograft and patient-derived xenograft model	50 mg/kg	Decreased tumour volume	[32]

**Table 3 cancers-13-00816-t003:** Clinical trials of HDAC inhibitors in gynaecological cancers.

NCT Number	Trial	Inhibitor Name	Inhibitor Type	Combination Agent(s)	Epigenetic Target	Epigenetic Inhibitor Dose	Cancer Type *	Recruitment Status
NCT04651127	Phase I/II	Chidamide	Class I HDAC inhibitor	Toripalimab	Class I HDACs	30 mg/day orally, twice a week	Persistent, Recurrent, or Metastatic Cervical Cancer	Recruiting
NCT02728492	Phase I	Quisinostat	HDAC inhibitor	Gemcitabine, Carboplatin, Paclitaxel	HDACs	8 mg every other day	Non-small Cell Lung Cancer, Epithelial Ovarian Cancer	Completed
NCT02948075	Phase II	Quisinostat	HDAC inhibitor	Carboplatin, Paclitaxel	HDACs	12 mg every other day	Ovarian Cancer	Completed
NCT02915523	Phase I/II	Entinostat	HDAC inhibitor	Avelumab	HDACs	5 mg weekly, 3 months	Advanced Epithelial Ovarian Cancer	Unknown
NCT00772798	Phase II	Vorinostat	HDAC inhibitor	Paclitaxel, Carboplatin	HDACs	400 mg once daily orally, days 4–10 of a 25 day cycle	Recurrent Ovarian Cancer	Unknown
NCT03345485	Phase I/II	Tinostamustine	alkylating HDAC inhibitor	-	HDACs	60 mg/m^2^ up to 100 mg/m^2^, day 1 and 15 of a 28 day cycle	Advanced Solid Tumors	Recruiting
NCT00020579	Phase I	Entinostat	HDAC inhibitor	-	HDACs	Dose escalation study	Advanced Solid Tumors or Lymphoma	Completed
NCT00421889	Phase I/II	Belinostat	HDAC inhibitor	Carboplatin, Paclitaxel	HDACs	1000 mg/m^2^, days 1–5 of a 21 day cycle	Ovarian Cancer in Need of Relapse Treatment	Completed
NCT00976183	Phase I/II	Vorinostat	HDAC inhibitor	Carboplatin, Paclitaxel	HDACs	200 mg once a day	Advanced Stage Ovarian Carcinoma	Terminated
NCT04315233	Phase I	Belinostat	HDAC inhibitor	Ribociclib	HDACs	600 mg/m^2^, days 1–5 of a 28 day cycle	Metastatic Triple Neg Breast Cancer & Recurrent Ovarian Cancer	Recruiting
NCT02601937	Phase I	Tazemetostat	KMT inhibitor	-	EZH2	Dose escalation study	Pediatric Relapsed or Refractory INI1-Negative Tumors or Synovial Sarcoma	Recruiting
NCT00301756	Phase II	Belinostat	HDAC inhibitor	-	HDACs	-	Recurrent or Persistent Ovarian Epithelial or Primary Peritoneal Cavity Cancer	Completed
NCT04703920	Phase I	Belinostat	HDAC inhibitor	Talazoparib	HDACs	Up to 1000 mg/m^2^ IV once daily on days 1– 5 of a 21-day cycle	Metastatic Castration Resistant Prostate Cancer, and Metastatic Ovarian Cancer	Not yet recruiting
NCT03018249	Phase I	Entinostat	HDAC inhibitor	Medroxyprogesterone Acetate	HDACs	-	Endometrial cancer	Active, not recruiting
NCT00132067	Phase II	Vorinostat	HDAC inhibitor	-	HDACs	-	Recurrent or Persistent Ovarian Epithelial or Primary Peritoneal Cavity Cancer	Completed
NCT02661815	Phase I	Ricolinostat	HDAC inhibitor	Paclitaxel	HDACs	80 mg/m2 per week (3 out of 4 weeks).	Gynecologic cancers	Terminated
NCT04357873	Phase II	Vorinostat	HDAC inhibitor	Pembrolizumab	HDACs	400 mg once daily, until progression	metastatic squamous cell carcinoma (head and neck, lung, cervix, vulva, anus and penis)	Recruiting
NCT04498520	Phase I	Abexinostat	HDAC inhibitor	Palbociclib, Fulvestrant	HDACs	-	Breast and Gynecologic cancers	Not yet recruiting
NCT00413322	Phase I	Belinostat	HDAC inhibitor	5-Fluorouracil	HDACs	300, 600, or 1000 mg/m^2^ belinostat for 5 days every 21 days starting with cycle 1	Advanced Solid Tumors	Completed

* may include other cancers in addition to gynaecological cancers.

## Data Availability

Data sharing not applicable. No new data were created or analyzed in this study. Data sharing is not applicable to this article.

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
