# Peer review of "Histone Modifying Enzymes in Gynaecological Cancers"

_cancers, 2021, doi:10.3390/cancers13040816_

Round 1

Reviewer 1 Report

The authors have discussed the involvement of histone-modifying enzymes as epigenetic modifiers in gynecological cancers. The article's title is misleading as it does not include all the epigenetic modifiers such as DNMTs and TETs. The authors should either discuss DNMTs and TETs in the epigenetic regulation of gynecological cancers or change the title accordingly. The authors have mentioned DNA methylation in their introduction, however, the article is focused on histone modifications.

In the epigenetic treatments section, the authors are mainly discussing the clinical trials of histone modifiers in other cancers and not in gynecological cancers.  This section should be primarily focused on gynecological cancers. Again, any preclinical studies or clinical trials related to the use of DNMT/TET inhibitors in gynecological cancers are missing. It will be great if authors could prepare a table of preclinical studies utilizing epigenetic modifiers. Please indicate the preclinical models, dose, timing of treatment, and the outcome, along with the references.

Also, prepare another table to indicate clinical trials of epigenetic modifiers in gynecological cancers. In the table, please mention the name of the drug, its target, dosage regime, type of gynecological cancer, and outcome (if the trial is finished)

Please include references in table 1.

Author Response

Comment/question from Reviewer - The authors have discussed the involvement of histone-modifying enzymes as epigenetic modifiers in gynecological cancers. The article's title is misleading as it does not include all the epigenetic modifiers such as DNMTs and TETs. The authors should either discuss DNMTs and TETs in the epigenetic regulation of gynecological cancers or change the title accordingly. The authors have mentioned DNA methylation in their introduction, however, the article is focused on histone modifications.

Response - We have altered the title to better capture the scope of the review. The new title is: “Histone Modifying Enzymes in Gynaecological Cancers”

We have also made added text in the background to highlight the review will specifically focus on histone modification (changes to text are underlined, line 61-64):

“This review will outline and discuss studies pertaining to epigenetic factors with a focus on histone modifying enzymes in common gynaecological cancers, specifically, ovarian, endometrial and cervical cancers. The mechanisms by which epigenetics contribute to tumorigenesis and the evidence that implicates epigenetic enzymes, in particular histone modifying enzymes, as treatment targets will be discussed.”

Comment/question from Reviewer - In the epigenetic treatments section, the authors are mainly discussing the clinical trials of histone modifiers in other cancers and not in gynecological cancers.  This section should be primarily focused on gynecological cancers.

Response - We have made changes throughout the text in Section 5 “Epigenetic Treatments in Gynaecological Cancers”. This section describes the clinical trials and treatments opportunities in of histone modifiers, and we have modified the text to better emphasize gynaecological cancers. Some of these text changes have been to remove trials in other cancer types, other changes have been to emphasise when we are referring to testing epigenetic inhibitors in models of gynaecological cancers.

We have also include a new table that summaries the clinical trials for HDACis in gynaecological cancers that are present on ClinicalTrials.gov (new Table 3).

Comment/question from Reviewer - Again, any preclinical studies or clinical trials related to the use of DNMT/TET inhibitors in gynecological cancers are missing.

Response - This is a similar comment to the first comment from Reviewer 1. As we have changed the title to better reflect the scope of the review (specifying histone modifying enzymes, as opposed to DNMTs and TETs) we believe this request by reviewer 1 is no longer applicable.    

Comment/question from Reviewer - It will be great if authors could prepare a table of preclinical studies utilizing epigenetic modifiers. Please indicate the preclinical models, dose, timing of treatment, and the outcome, along with the references.

Response - We have added a new Table 2 to summarize some of the preclinical studies and have included the preclinical models used, treatment doses, the outcome, and any references. Please note this is not an exhaustive list, as the literature is broad and we feel a systematic review of preclinical studies is out of scope for this review. However the inclusion of this Table does highlight an example of preclinical studies that have been undertaken.

To include Table 2 we have added the following text to the manuscript (line 370-372):

“There have been a variety of preclinical studies testing epigenetic inhibitors in models of gynaecological cancers, some of which are highlighted in this review and Table 2, however this is not an exhaustive list as only representative examples of recent studies have been included.”

Comment/question from Reviewer - Also, prepare another table to indicate clinical trials of epigenetic modifiers in gynecological cancers. In the table, please mention the name of the drug, its target, dosage regime, type of gynecological cancer, and outcome (if the trial is finished)

Response - We are thankful for this suggestion as the inclusion of this Table has improved the manuscript and ensures a focus on gynecological cancers. We have included a new Table 3 that mentions the name of each drug, its epigenetic target, any combination therapy, dosage regimen and type of gynecological cancer targeted. We did not include outcome in this table as it was generally not available for the trials, however we have already discussed some of the trial outcomes in the text.

To include Table 3 we have added the following text to the manuscript (line 370-372):

“A variety of HDAC inhibitors are currently or have been trialled within Phase I or II trials for gynaecological cancers, a summary of trials on ClinicalTrials.gov is provided in Table 3.”

Comment/question from Reviewer - Please include references in table 1.

Response - We have added a new column to Table 1 that includes references.

Reviewer 2 Report

The review is thorough and covered a wide range of information. The review will look complete with the addition of another section on “Combination Therapy”. The treatment part could be improved. Combination therapy describing the combination of epigenetic drugs with other drugs will be an excellent addition to this review. The authors in some places mentioned how epigenetic drugs sensitize drug resistance cancer cells. All these information could be organized in one section. The combination therapy is emerging and the following references are provided as suggestions to develop a section on combination therapy. The addition of this section will improve the quality of this well written review.

Rational Cancer Treatment Combinations: An Urgent Clinical Need

Boshuizen et al. Molecular Cell, Volume 78, ISSUE 6, P1002-1018, June 18, 2020

Leary et al. Sensitization of Drug Resistant Cancer Cells: A Matter of Combination Therapy. Cancers. 10, 483. 2018. https://doi.org/10.3390/cancers10120483

Author Response

Comment/question from Reviewer - The review is thorough and covered a wide range of information. The review will look complete with the addition of another section on “Combination Therapy”. The treatment part could be improved. Combination therapy describing the combination of epigenetic drugs with other drugs will be an excellent addition to this review. The authors in some places mentioned how epigenetic drugs sensitize drug resistance cancer cells. All these information could be organized in one section. The combination therapy is emerging and the following references are provided as suggestions to develop a section on combination therapy. The addition of this section will improve the quality of this well written review.

Rational Cancer Treatment Combinations: An Urgent Clinical Need

Boshuizen et al. Molecular Cell, Volume 78, ISSUE 6, P1002-1018, June 18, 2020

Leary et al. Sensitization of Drug Resistant Cancer Cells: A Matter of Combination Therapy. Cancers. 10, 483. 2018. https://doi.org/10.3390/cancers10120483

Response - We thank the reviewer and agree the addition of this section has improved the review. We have included a new section titled “5.4 Combination therapy” and have included the references suggested by the reviewer. The new text reads (line 507-523):

“The rationale for combination therapies in cancer treatment has been reviewed [117]. As alluded to throughout this review, combination therapy with various agents may augment the anticancer effect of epigenetic inhibitors [118,119]. For example, the combination of HDAC inhibitors and immunotherapy appears to be beneficial through transcriptional induction of endogenous retroviruses and testis antigens, resulting in a response called viral mimicry [118]. In vitro and in vivo xenograft studies in ovarian cancer cell lines and mouse models have shown that HDAC inhibitor treatment results in re-sensitisation of ovarian cancer cells to paclitaxel [120]. Additionally, the combination of PARP inhibitors with BET inhibitors resulted in increased DNA damage in ovarian cancer cells, potentially by blocking homologous recombination repair [121]. Moreover, epigenetic inhibitors can be combined with each other to induce cell death; for example, the combination of HDAC inhibition and demethylating agents such as 5-azacytidine and decitabine exerted synergistic effects in inhibiting proliferation of ovarian cancer cells [122]. Results from these studies illustrate the range of processes which can be modulated by epigenetic inhibition, and thus can be leveraged to enhance the cytotoxic activity of other agents. As such, more research into combination therapies with epigenetic inhibitors is well warranted.”

Reviewer 3 Report

This is very well written and interesting review with good coverage of the topic.

In this review, the authors begin the review with a background on epigenetics and factors that influence in the function. The epigenetic system is quite complex so explaining it in an easy way is difficult. The text in the background section is “hard stuff” for readers that are not molecular biologist, but description of a complex matter is not easy to make “easy reading”.
The following 3 sections on Epigenetic modifiers in ovarian, endometrial and cervical cancer and the following more general section on Epigenetic Treatments in Gynecologic Cancers are very helpful, even for clinicians.

One minor remark
A typing error in line 197. KDM family of also … of should be deleted.

Author Response

Comment/question from Reviewer - In this review, the authors begin the review with a background on epigenetics and factors that influence in the function. The epigenetic system is quite complex so explaining it in an easy way is difficult. The text in the background section is “hard stuff” for readers that are not molecular biologist, but description of a complex matter is not easy to make “easy reading”. The following 3 sections on Epigenetic modifiers in ovarian, endometrial and cervical cancer and the following more general section on Epigenetic Treatments in Gynecologic Cancers are very helpful, even for clinicians.

Response - We thank the reviewer for acknowledging this is a complex area, and we are pleased that some sections will be readable by a broader audience.

Comment/question from Reviewer - One minor remark
A typing error in line 197. KDM family of also … of should be deleted.

Response - This has been corrected. The text now reads: “Members of the KDM family also …..”

Round 2

Reviewer 1 Report

The changes made by the authors have significantly improved the manuscript.